# Heat Capacities of L-Cysteine, L-Serine, L-Threonine, L-Lysine, and L-Methionine

**DOI:** 10.3390/molecules28010451

**Published:** 2023-01-03

**Authors:** Václav Pokorný, Vojtěch Štejfa, Jakub Havlín, Michal Fulem, Květoslav Růžička

**Affiliations:** 1Department of Physical Chemistry, University of Chemistry and Technology, Prague, Technická 5, CZ-166 28 Prague, Czech Republic; 2Institute of Macromolecular Chemistry, Czech Academy of Sciences, Heyrovského nám. 2, CZ-162 06 Prague, Czech Republic; 3Central Laboratories, University of Chemistry and Technology, Prague, Technická 5, CZ-166 28 Prague, Czech Republic

**Keywords:** L-cysteine, L-serine, L-threonine, L-lysine, L-methionine, crystalline phase, heat capacity

## Abstract

In an effort to establish reliable thermodynamic data for amino acids, heat capacity and phase behavior are reported for L-cysteine (CAS RN: 52-90-4), L-serine (CAS RN: 56-45-1), L-threonine (CAS RN: 72-19-5), L-lysine (CAS RN: 56-87-1), and L-methionine (CAS RN: 63-68-3). Prior to heat capacity measurements, initial crystal structures were identified by X-ray powder diffraction, followed by a thorough investigation of the polymorphic behavior using differential scanning calorimetry in the temperature range from 183 K to the decomposition temperature determined by thermogravimetric analysis. Crystal heat capacities of all five amino acids were measured by Tian–Calvet calorimetry in the temperature interval (262–358) K and by power compensation DSC in the temperature interval from 215 K to over 420 K. Experimental values of this work were compared and combined with the literature data obtained with adiabatic calorimetry. Low-temperature heat capacities of L-threonine and L-lysine, for which no or limited literature data was available, were measured using the relaxation (heat pulse) calorimetry. As a result, reference heat capacities and thermodynamic functions for the crystalline phase from near 0 K to over 420 K were developed.

## 1. Introduction

This work is a continuation of our project, for which the goal is to establish reliable thermodynamic data along the saturation curve for a group of proteinogenic amino acids [1,2,3,4,5]. The five amino acids studied in this work (L-cysteine, L-serine, L-threonine, L-lysine, L-methionine) are white crystalline compounds under ambient conditions and are reported to decompose at/prior to melting in the temperature range 523–573 K [6]. 

L-threonine, L-lysine, and L-methionine are essential for humans, and L-cysteine is essential for children. L-serine [7] is traditionally considered a non-essential amino acid, but it has been found that vertebrates are unable to synthesize optimal amounts of it throughout their lives under certain circumstances. Amino acids are also used as excipients to stabilize the amorphous form of drugs in formulations aimed at enhancing the dissolution and oral bioavailability of poorly water-soluble drugs [8]. Note that L-lysine is often used in the form of sulfide or hydrochloride due to its low stability and high hygroscopicity [9,10].

Significance and many other aspects of the title, as well as other amino acids, are summarized in reviews [6,11]. Despite their wide use, only a few thermodynamic datasets are available for them in the region of technologically important temperatures. In this study, experimental crystal heat capacities for the title compounds are reported along with derived thermodynamic properties in the temperature range near 0 K to over 420 K. After establishing the crystal structures for all amino acids studied by X-ray powder diffraction (XRPD), the phase behavior was studied using heat flux differential scanning calorimetry (HF DSC) in order to detect possible phase transitions. The decomposition temperatures were determined by thermogravimetry. Heat capacities in the temperature range of 260 –350 K were determined with Tian-Calvet calorimetry and extended down to 215 K and up to 420 K or higher using power compensation DSC. Literature low-temperature heat capacities (adiabatic calorimetry from ca 10 K to ca 300 K) were found for four compounds (all except for L-lysine). For L-lysine and L-threonine (for which literature adiabatic data [12] are questionable), new measurements by means of thermal-relaxation calorimetry (2 –300 K) were performed.

## 2. Results and Discussion

### 2.1. Thermogravimetric Analysis (TGA)

Traditionally, amino acids are reported to melt or decompose at temperatures above 450 K [6,11]. However, literature data exhibits a significant scatter. Moreover, a recent paper by Weiss et al. [13] suggests that all amino acids studied do not melt but rather decompose, which could damage calorimeters as a result. Weiss et al. [13] reported an evolution of CO_2_, NH_3_, and H_2_O during the decomposition of L-cysteine at the temperature of about 518 K. For the remaining four amino acids of this study, Weiss et al. [13] reported the decomposition without specifying its temperature and mechanism. Therefore, reinvestigation of the melting/decomposition process for all five amino acids was undertaken to reconcile the literature data. All the samples were studied by TGA; results are shown in Figure 1.

The decomposition behavior differs for particular amino acids. The simplest decomposition path can be seen in the case of L-cysteine (Figure 1a), L-threonine (Figure 1c), and L-methionine (Figure 1e), which show a one-stage decomposition. For L-serine (Figure 1b), two decomposition stages can be clearly distinguished. L-lysine (Figure 1d) exhibits the most complicated decomposition behavior. We can see two distinct decomposition stages, but the first decrease in sample weight appears already around 300 K and also around 395 K, both accompanied by wide DSC peaks. They most likely correspond to the evaporation of water from the hygroscopic sample, as exposure to air cannot be avoided during the preparation of the sample for the TGA measurements. The weight fraction before the main decomposition peak is around 0.97, meaning water accounts for approximately 3 % of the total sample weight. It is known that L-lysine absorbs air moisture and forms hemihydrate easily [9], which should contain 5 % of water in weight.

One thermal event was observed for L-cysteine around 445 K (i.e., shortly below the decomposition temperature) and two in L-methionine (at 393 K and 415 K). In the case of L-lysine, another sharp peak at 371 K was also observed. Any of these peaks are not accompanied by weight loss, and thus they correspond to phase transitions. A more detailed analysis of the phase behavior was performed using HF DSC, as described in Section 2.2. 

A summary of decomposition temperatures obtained in this work, as well as those reported in the literature, is presented in Table 1. It should be noted that the decomposition kinetics is influenced by a number of factors (heating rate, sample size, purge gas, etc.); this is reflected by a relatively large range of reported decomposition temperatures. Recently reported temperatures of fusion obtained with fast scanning DSC [14] (see Table 1) are higher than those of decomposition by 50–140 K. To avoid the decomposition, Do et al. [14] used scanning rates up to 20,000 K·s^−1^.

The agreement of our TGA data with the literature is only partial. TGA data by Rodante et al. [15] for L-cysteine, L-serine, and L-threonine lie within 6 K from our data. For L-lysine and L-methionine, the differences are very large (152 K and 38 K, respectively). In the case of L-lysine, the low “decomposition temperature” value reported by Rodante et al. [15] likely corresponds to the evaporation of water from a wet sample, as also observed in our experiments. In the case of L-methionine, the reason for this discrepancy is unclear. When comparing to another set of TGA data by Rodriguez-Mendez et al. [16], there is again a reasonable agreement for three of the amino acids (decomposition temperatures of L-cysteine, L-serine, and L-lysine fall within 20 K from our values); however, their data for L-threonine and L-methionine are significantly lower (by 38 K and 49 K, respectively). If we compare to data recommended by the Syracuse Research Corporation (retrieved from [17]), there is an agreement within 10 K for four of the compounds (this time, the decomposition temperature for L-cysteine is in poor agreement). Agreement with other data obtained by DSC, differential thermal analysis (DTA), or differential thermal gravimetry (DTG) is similarly inconsistent.

**Table 1 molecules-28-00451-t001:** Temperatures of Fusion and Decomposition of L-Cysteine, L-Serine, L-Threonine, L-Lysine, and L-Methionine ^a^.

Reference	*T*_decomp, onset_/K	*T*_decomp, peak top_/K	Method	Scanning Rate, Purge Gas
L-cysteine (one-stage decomposition) (*T*_fus_= 604 ± 7 K ^b^)				
SRC recommendation ^c^	513			
Weiss et al. [13]		518	TGA	5 K min^−1^, argon
Olafsson and Bryan [18]		521	DSC	10 K min^−1^, nitrogen
Rodriguez-Mendez et al. [16]	493		TGA	10 K min^−1^, air
Rodriguez-Mendez et al. [16]	496		DTA	10 K min^−1^, air
Rodante et al. [15]	393,441,490 ^d^		TGA	10 K min^−1^, nitrogen
Rodante et al. [15]		415,453,521	DSC	10 K min^−1^, nitrogen
Wesolowski and Erecinska [19]		473	DTA	5 K min^−1^, air
Wesolowski and Erecinska [19]		473	DTG	5 K min^−1^, air
This work	484		TGA	5 K min^−1^, argon
L-serine (two-stage decomposition) (*T*_fus_= 519 ± 7 K ^b^)				
SRC recommendation ^c^	501			
Olafsson and Bryan [18]		505	DSC	10 K min^−1^, nitrogen
Rodriguez-Mendez et al. [16]	503		TGA	10 K min^−1^, air
Rodriguez-Mendez et al. [16]	502		DTA	10 K min^−1^, air
Rodante et al. [20]	496		TGA	10 K min^−1^, nitrogen
Rodante et al. [20,21]		508	DSC	10 K min^−1^, nitrogen
This work	491		TGA	5 K min^−1^, argon
L-threonine (one-stage decomposition) (*T*_fus_= 587 ± 9 K ^b^)				
SRC recommendation ^c^	529			
Olafsson and Bryan [18]		532	DSC	10 K min^−1^, nitrogen
Rodriguez-Mendez et al. [16]	488		TGA	10 K min^−1^, air
Rodriguez-Mendez et al. [16]	471		DTA	10 K min^−1^, air
Rodante et al. [20]	520		TGA	10 K min^−1^, nitrogen
Rodante et al. [20,22]		540	DSC	10 K min^−1^, nitrogen
Contineanu et al. [23]	521	529	DSC	4 K min^−1^, nosp.
Wesolowski and Erecinska [19]		523	DTA	5 K min^−1^, air
Wesolowski and Erecinska [19]		518	DTG	5 K min^−1^, air
This work	526		TGA	5 K min^−1^, argon
L-lysine (two-stage decomposition) (*T*_fus_= 529 ± 9 K ^b^)				
SRC recommendation ^c^	498			
Olafsson and Bryan [18]		506	DSC	10 K min^−1^, nitrogen
Rodriguez-Mendez et al. [16]	509		TGA	10 K min^−1^, air
Rodriguez-Mendez et al. [16]	531		DTA	10 K min^−1^, air
Rodante et al. [20]	336		TGA	10 K min^−1^, nitrogen
Rodante et al. [20]		339,515,532	DSC	10 K min^−1^, nitrogen
This work	492		TGA	5 K min^−1^, argon
L-methionine (one-stage decomposition) (*T*_fus_= n.a. ^e^)				
SRC recommendation ^c^	555			
Olafsson and Bryan [18]		562	DSC	10 K min^−1^, nitrogen
Rodriguez-Mendez et al. [16]	498		TGA	10 K min^−1^, air
Rodriguez-Mendez et al. [16]	467		DTA	10 K min^−1^, air
Rodante et al. [22]	509 ^d^		TGA	10 K min^−1^, nitrogen
Rodante et al. [22]		568	DSC	10 K min^−1^, nitrogen
This work	547		TGA	5 K min^−1^, argon

^a^ Sources where melting/decomposition temperature is merely mentioned are not listed. Values in this table are rounded to the nearest kelvin. ^b^ Fast scanning DSC was used by Do et al. [14] for *T*_fus_ determination (see text). ^c^ “Temperature of fusion” (in fact, *T*_decomp_) recommended by the Syracuse Research Corporation (SRC) and used in the past by many authors (retrieved from [17]). ^d^ Start of the TGA peak (onset not reported). ^e^ Do et al. [14] were not able to determine the temperature of fusion due to the overlap of melting and evaporation processes.

### 2.2. Phase Behavior

All amino acids studied are zwitterionic crystals at 298.15 K whose structures determined by XRPD are provided in Table 2. Subsequently, the phase behavior was investigated in the temperature range from 183 K to the thermal decomposition temperature using heat-flux DSC to confirm/exclude the presence of phase transitions. The phase transitions observed in this work are listed and compared with literature values in Table 3.

Two ambient-pressure polymorphs of L-cysteine are usually considered in the literature: I (orthorhombic) [37], which exhibits a phase transition at 70 K [24] and should be thus considered two phases (Ia and Ib), and II (monoclinic) [38]. We are not aware that their thermodynamic relationship was resolved. Additionally, two high-pressure polymorphs, III and IV, were described [39]. The thermal event observed with TGA near 450 K (see Figure 1a) was confirmed using HF DSC, as shown in Figure 2a. This phase transition could represent a transition from I_b_ to II; in that case, the phases would be enantiotropically related with form II stable above approximately 400 K. Detailed study of the phase transition is complicated because of thermal decomposition and identity of high-temperature phase with crystal structure II could not be confirmed.

For L-serine, four polymorphs are known to date. The only polymorph stable at normal pressure is I (sometimes also labeled 1). The other polymorphs were reported to exist at high pressures: II [40], III [41], and IV [42]. In addition to a single known ambient-pressure polymorph I of L-threonine, three high-pressure polymorphs were reported: I’, II, and III [43]. No phase transition for L-serine and L-threonine has been observed in this work in the studied temperature range.

Since L-lysine forms hemihydrate and monohydrate in a moist atmosphere, no studies of the phase behavior of anhydrate were reported to our knowledge. Multiple phase transitions have been detected for L-lysine in this work, which can be seen on the respective DSC curves in Figure 2b. During the first heating of L-lysine hemihydrate in a pierced DSC pan, a wide peak corresponding to the evaporation of water is observed. In a subsequent run, L-lysine anhydrate exhibits a wide peak with a maximum of 292 K, which corresponds to a second-order transition, and a sharp peak of a first-order phase transition at 370.3 ± 0.5 K.

Three structures of L-methionine (II, III, and IV), all with disordered side chains, were recently solved [29]. On the other hand, previous crystal structure reports at lower [44,45] and room [28] temperatures did not reveal any disorder. Solving the structure of high-temperature phase I was not attempted because of the complicated disorder of phase II [29]. The structures resolved at 120 K [44], 293 K [29], and 320 K [29] are all monoclinic; they have very close unit-cell parameters and differ only in occupancies of the disordered side chain (e.g., molecule B has occupancy 1:0:0:0 [44], 0.69:0.12:0.08:0.11 [29], and 0.41:0.15:0.20:0.24 [29], respectively). However, the previous calorimetric studies [34,36], as well as this work, revealed only one phase transition in the temperature region 120– 320 K manifested by a very broad peak with a start around 250 K and a maximum at 307. K. Lima Jr. et al. [31] reported another anomaly in the intensities of some Raman bands in the interval 160–140 K and around 220 K during heat capacity measurements “by relaxation calorimetry (note that the paper does not tabulate any heat capacity data) but not in their DSC experiments. The anomalies scattered over wide temperature ranges are said to be related to conformational changes of at least one of the independent molecules in the unit cell [31]. Based on an unsubstantiated criterion Δ_tr_*S* > 1 J mol^−1^ K^−1^, the phase transition with a maximum at 307 K is then established as a first-order transition by authors [31].

The tangled story of the L-methionine phase behavior below 350 K has a consistent and logical solution, although somewhat uncommon. The shape and negligible hysteresis of the peak corresponding to the phase transition suggests that it might be a lambda transition, i.e., a second-order transition, where the continuous structural changes intensify while increasing the temperature and suddenly decelerate or stop after reaching a certain point. From the calorimetric point of view, the intensification of the structural changes manifests through a “divergence” of heat capacity followed by a fast drop to a value following the trend well below the temperature region of the structural changes (see Figure 2c). At cooling, this lambda transition shows almost no hysteresis—the heat capacity rises very fast, reaches the maximum at the same temperature, and then slowly decays. The temperature of the maximum at the heat capacity curve (about 308 K) would be considered as the phase transition temperature (here between phases IV and III) since any other characteristic temperature cannot be evaluated for the transition with similar accuracy. 

From the crystallographic point of view, the situation is somewhat surprising. Polymorph IV exhibits a continuous (and equilibrium) change of the occupancies of conformations, starting from a fully ordered structure at low temperatures all the way until reaching occupancies corresponding to polymorph III. Initially, the occupancies of alternative conformations are below the detection limit, but with rising temperature, the rise of their occupancies intensifies, and once it reaches values for polymorph III, they remain stable. In other words, disorder occupancies of polymorph IV may match any values between ordered structure and values for polymorph III, depending on the temperature. A variable-temperature x-ray study should confirm this hypothesis, but its execution is very demanding because of the large number of occurring conformations.

Although the explained behavior differs from the common idea of polymorphism, it is consistent and logical. One might argue that the moment where the disorder starts to appear should be considered a phase transition instead of the moment when the occupancies stabilize. However, because of the diverging nature of the process, it is not possible to determine its beginning while its end is well-defined. It is vital to watch the behavior of the first derivatives of the thermodynamic and structural properties in the case of second-order phase transitions. While the thermodynamic properties and structural parameters are continuous, their first derivatives are discontinued—and despite the limited amount of structural data, calorimetric experiments clearly place this discontinuity at 308 K. 

An interesting link between the crystallographic and calorimetric observations can be found that supports the hypothesis above. If the total entropy change Δ_IV-III_*S* is evaluated from the heat capacity description (Section 2.3), a value of 11.4 J mol^−1^ K^−1^ is obtained. Although a somewhat arbitrary baseline needs to be selected, this value is in good agreement with a mixing entropy corresponding to the disorder of molecule B in the crystal structure of form III calculated from occupancies reported in [29], Δ_mix_*S* = 10.9 J mol^−1^ K^−1^. Disorder of molecule A would correspond to additional Δ_mix_*S* = 5.2 J mol^−1^ K^−1^, but since the two reported conformations [29] are noticeably similar, the mixing entropy might be an irrational construct in this case. Agreement between calorimetric Δ_IV-III_*S* and crystallographic Δ_mix_*S* supports the hypothesis that the wide calorimetric peak corresponds to a continuous transformation from the fully ordered form IV to form III. The Raman spectroscopic experiment [31] seems to be in agreement with this hypothesis, while the anomaly in relaxation calorimetry heat capacity measurements [31] might be related to the metastable occurrence of fully ordered structure at elevated temperatures and relaxation to equilibrium disordered form IV. 

The discussion of the other phase transitions of L-methionine visible in Figure 2c is simpler. Phase transition III-II is sharp, exhibits hysteresis, and its shape differs on heating and on cooling. Structures are determined at a single temperature for both polymorphs [29] and show that new conformations are introduced during the phase transition. This description agrees with typical features of a first-order phase transition. Phase transition II-I is associated with the smallest entropy change, and it is difficult to reveal the shape of its calorimetric peak without ambiguity. Resolving the structure of form I was not attempted in [29] because of expecting a complex disorder. The choice of considering the phase transition to be of the first or second order brings negligible difference to the description of thermodynamic quantities.

### 2.3. Heat Capacities

Experimental heat capacities obtained in this work with two Tian-Calvet calorimeters (SETARAM μDSC IIIa, SETARAM MicroCalvet), PerkinElmer power compensation DSC 8500 (PC DSC), and Quantum Design PPMS relaxation calorimeter are listed in the Appendix A including correction scaling factors applied for PerkinElmer DSC 8500 and Quantum Design PPMS results. Available literature data on crystal heat capacities are summarized in Table 4. Experimental data from Table 4 were fitted with Equations (1) and (2) parameters which are given in Table 5.

The experimental heat capacities for all amino acids studied are compared with the smoothed values obtained using Equations (1) and (2) in Figure 3. The deviations of the selected experimental data (marked bold in Table 4) from the smoothed values do not exceed 1 % with the exception of lowest temperatures, where all experiments have higher uncertainty, and vicinity of phase transitions, where the data points were excluded from the fit. 

Adiabatic data for L-cysteine by Huffman and Ellis [46] have a very different slope from data by Paukov et al. [30] and data obtained in this work by both Tian-Calvet and power compensation DSC (deviations from −8 to +13%). This was commented on by Paukov et al. [30], who believe that this discrepancy is caused by impurities (possibly DL form or monoclinic L form of cysteine) in the sample of Huffman and Ellis [46]. In the high-temperature region, heat capacities of L-cysteine were only correlated below 435 K because of discrepancies in experimental data above this temperature, which probably correspond with the initiation of the suggested I_b_-II phase transition observed during the phase behavior study. Adiabatic heat capacity data by Hutchens et al. [47] is available for L-serine. Our Tian-Calvet and PC DSC data agree with theirs within 1%. Data by Lukyanova et al. [12] for L-threonine is also obtained by adiabatic calorimetry [49]. The stated uncertainty of 0.2 % [12] is, however, questionable, when comparing data from the same laboratory [50,51] with previously published data [4,52,53] (for more details see Appendix A and Section 1 of the Appendix A). Although not included in the correlation, most of the data points from [12] deviate from our fit (based on our measurements using Tian-Calvet, PC DSC, and relaxation calorimetry) by less than 2%. Heat capacities of L-lysine obtained from the three calorimeters are in good mutual agreement, and all exhibit a wide peak with a maximum of around 291 K, which was found to be characteristic of anhydrous L-lysine during the phase-behavior study (see Figure 2). This transition, as well as the one exhibited by L-methionine around 310 K, is treated as a lambda transition in the correlation. I.e., heat capacities of both phases in the vicinity of the phase transition are correlated separately, and the phase transition temperatures in Table 3 are evaluated as the intersection of the extrapolated heat capacity curves. Consequently, heat capacity, as well as entropy and enthalpy, are continuous at these phase transitions. For the other phase transitions of L-methionine (at 393 K and 421 K) and L-lysine (at 370.3 K), a step change of enthalpy equal to the value obtained by DSC (Table 3) is considered in the correlation. Heat capacities of the given phases are correlated separately except for the crII and crI phases of L-methionine, which were treated together since only two experimental data points were available for crII. From the heat capacity data for L-methionine by Hutchens et al. [34], only those from the low-temperature calorimeter were used, which are in reasonable agreement with our Tian-Calvet and PC DSC data (within 1.5%), although they show some scatter when approaching the crIV-crIII phase transition. Heat capacity data from the high-temperature adiabatic calorimeter are related with a higher uncertainty according to the authors [34], their scatter is noticeably high and agreement with our measurements for crIII is poor.

The thermodynamic functions of L-cysteine, L-serine, L-threonine, L-lysine, and L-methionine obtained using Equations (1) and (2) are tabulated in Appendix A and shown graphically in Figure 4. Their values at *T* = 298.15 K are shown in Table 6 for convenience. Note that L-lysine and L-methionine are in close proximity to a second-order phase transition at *T* = 298.15 K, leading to an increased heat capacity.

## 3. Materials and Methods

### 3.1. Samples description

The title amino acids were of commercial origin; they were used as received, with the exception of L-lysine, which was dried for heat capacity and phase behavior measurements. L-lysine drying was performed for 30 days at 343 K and 50 Pa for Tian-Clavet calorimetry and in situ by heating a pierced pan to at least 393 K for DSC measurements. The sample purities (as stated in the certificate of analysis provided by the manufacturer) are reported in Table 7.

### 3.2. Thermogravimetry 

Thermogravimetric analysis (TGA) was performed prior to heat capacity and phase behavior measurements. The thermogravimetric analyzer SETARAM Setsys Evolution was used. All samples were placed in an open platinum 100 μL crucible employing the temperature range (298 to 573) K with a temperature gradient of 5 K min^−1^ under an inert Ar atmosphere. 

### 3.3. Phase Behavior Measurements

XRPD was used to characterize initial crystal structures of the amino acids studied using a *θ*–*θ* powder diffractometer X’Pert3 Powder from PANalytical in Bragg-Brentano para-focusing geometry using wavelength CuKα radiation (*λ* = 1.5418 Å, *U* = 40 kV, *I* = 30 mA). The samples were scanned at 298.15 K in the range of 5°–50° 2*θ* with a step size of 0.039° 2*θ* and 0.7 s for each step. The diffractograms were analyzed with the software HighScore Plus in combination with yearly updated powder diffraction databases PDF4+ and PDF4/Organics.

The heat flux DSC TA Q1000 was used for the investigation of the phase behavior of the amino acids studied in the temperature range from 183 K to the decomposition temperature. The combined expanded uncertainty (0.95 level of confidence) in the phase transition temperatures and enthalpies are listed in Table 3. 

### 3.4. Heat Capacity Measurements

A Tian-Calvet type calorimeter (SETARAM μDSC IIIa) was used for the measurement of heat capacities in the temperature range from 266 –353 K for all compounds except L-lysine, for which another Tian-Calvet type calorimeter (SETARAM MicroCalvet) was used in the temperature range from 232 –360 K. As the detailed description of the calorimeters and their calibration and operation is reported in other works [1,54], only the most salient information is provided here. The heat capacity measurements were carried out by the continuous heating method [55], using the three-step methodology, i.e., the measurement of the sample is followed by the measurement of reference material (synthetic sapphire, NIST Standard reference material No. 720) and by performing a blank experiment. The saturated molar heat capacities $C_{\mathrm{sat}}$ obtained in this work are identical to isobaric molar heat capacities $C_{pm}^{o}$ in the temperature range studied, given the very low sublimation pressure of the samples. The combined expanded uncertainty (0.95 level of confidence) of the heat capacity measurements is estimated to be $U_{c}(C_{pm}^{o})=0.01\cdot C_{pm}^{o}$.

The PerkinElmer power compensation DSC 8500 equipped with an autosampler was used for the heat capacity determination in the temperature range of 215–470 K. For all studied compounds, with the exception of L-methionine, the upper temperature was lowered to avoid decomposition. The heat capacity measurements were carried out by the temperature increment method, repeated 3 times to eliminate systematic errors. The combined expanded uncertainty (0.95 level of confidence) of the heat capacity measurement is estimated to be $U_{c}(C_{pm}^{o})=0.03\cdot C_{pm}^{o}$. Due to lower accuracy, results from this calorimeter were slightly adjusted to agree with results from the more accurate SETARAM μDSC IIIa, following the common practice [56]. Scaling factors are presented in tables containing experimental heat capacities (see Appendix A), and maximum correction amounted to 0.01$\cdot C_{pm}^{o}$ in the case of L-lysine. The same procedure was also applied for low-temperature relaxation calorimetry, described in the next paragraph (see Appendix A).

For low-temperature heat capacity measurements of L-threonine and L-lysine, commercially available apparatus Physical Property Measurement System (PPMS) Model 6000 EverCool II (Quantum Design, San Diego, USA) equipped with heat capacity module (^4^He, *T*_min_ = 1.8 K) was used. The calorimeter uses a thermal-relaxation measurement technique which is an alternative to the time and labor-intensive adiabatic calorimetry. The specific heat capacity of a sample is determined by measuring the thermal response to a change in heating conditions [57]. Samples were placed in Cu cups (height 3.5 mm, diameter 3 mm) made from 0.025 mm thick copper foil (Alfa Aesar, purity: mass fraction purity 0.99999) by a technique similar to Shi et al. [58]. In contrast to Shi et al. [58], samples were not mixed with Apiezon N; instead, samples enclosed in a Cu cup were pressed to a height of ca 1 mm using stainless steel die (Maassen, Germany) and a press (Trystom, Czech Republic) using a force of 15 kN. The heat capacity of the Cu cup was subtracted from the total heat capacity using data recommended by Arblaster [59]. This technique was checked by measurement of compounds with reliable data obtained by adiabatic calorimetry (anthracene [60], L-asparagine [61], and glycine [62]), and uncertainty of results was found comparable to this reported previously [63] for samples encapsulated in Al DSC pans. The combined expanded uncertainty (0.95 level of confidence) of the heat capacity measurements is estimated to be $U_{c}(C_{pm}^{o})=0.10\cdot C_{pm}^{o}$ below 10 K, $U_{c}(C_{pm}^{o})=0.03\cdot C_{pm}^{o}$ in the temperature range of 10–40 K, and $U_{c}(C_{pm}^{o})=0.02\cdot C_{pm}^{o}$ in the temperature range of 40–300 K.

To describe the temperature dependence of heat capacity $C_{pm}^{o}$ in a wide temperature range (including literature heat capacities obtained using adiabatic calorimetry and data obtained by Quantum Design PPMS), the following equation proposed by Archer [64] was used: (1)$$C_{pm}^{o}/C_{pm}^{\mathrm{ref}}={\left(\frac{T}{T^{\mathrm{ref}}f(T)+bT}\right)}^{3}$$
where *T* is thermodynamic temperature, $T^{\mathrm{ref}}$= 1 K, and $C_{pm}^{\mathrm{ref}}$=1 J∙K^−1^∙mol^−1^ and
(2)$$f(T)=a_{i}{\left(T-T_{i}\right)}^{3}+b_{i}{\left(T-T_{i}\right)}^{2}+c_{i}\left(T-T_{i}\right)+d_{i}$$
where *a_i_*, *b_i_*, *c_i_* and *d_i_* are adjustable parameters, of which only a single parameter (*d_i_*) per each temperature interval is to be optimized while the values of the other three are imposed by a constraint of continuity and smoothness of the resulting temperature dependence. Parameter *b* is estimated from the slope of f(T) at temperatures greater than 70 K prior to the optimization procedure [64].

## 4. Conclusions

The decomposition temperatures of the five studied amino acids were determined by means of TGA, all in the range of 484–573 K (see Figure 1 and Table 1). The phase behavior was studied with heat flux differential scanning calorimetry in the temperature range from 183 K to the decomposition temperature to determine the presence of possible phase transitions. Multiple transitions have been detected for L-lysine and L-methionine. New heat capacity data have been obtained for L-cysteine, L-serine, L-threonine, L-lysine, and L-methionine using a combination of three different calorimetric techniques (relaxation, Tian-Calvet, PC DSC). Accurate equations for crystal heat capacity in the temperature range from 0–420 K or higher were established. These equations are based on new heat capacity measurements performed in this work as well as on low-temperature adiabatic heat capacity data from the literature. Standard thermodynamic functions (entropy, enthalpy, and Gibbs energy) of the crystalline phase at *p* = 0.1 MPa were evaluated in the temperature range from 0–420 K or higher (see Appendix A).

## Figures and Tables

**Figure 1 molecules-28-00451-f001:**
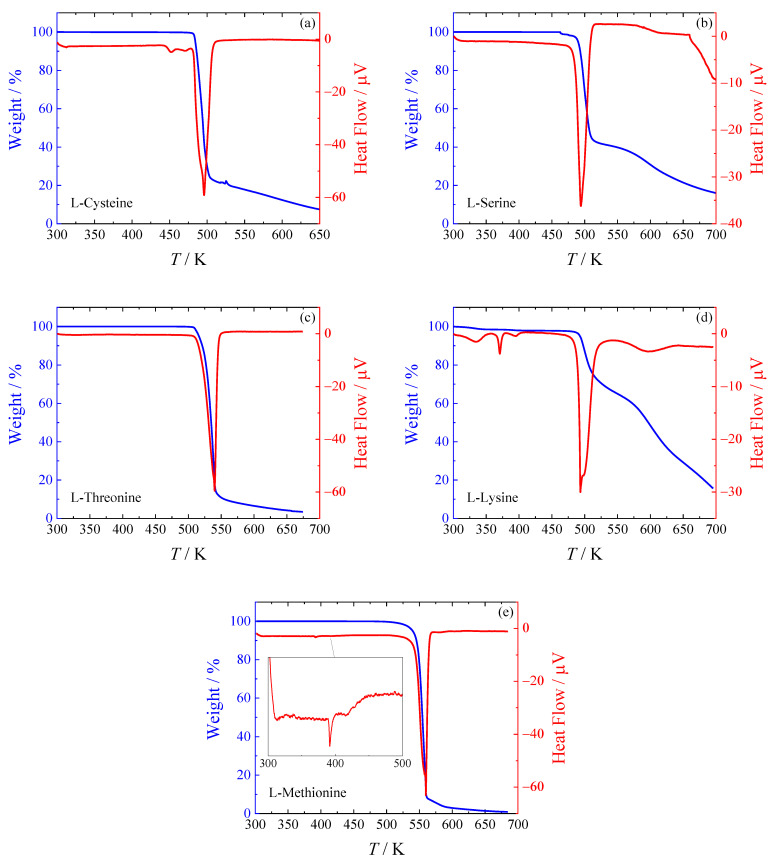
TGA analysis of studied amino acids. (**a**) L-cysteine, (**b**) L-serine, (**c**) L-threonine, (**d**) L-lysine, (**e**) L-methionine (with an inset detailing the heat flow at temperatures before decomposition).

**Figure 2 molecules-28-00451-f002:**
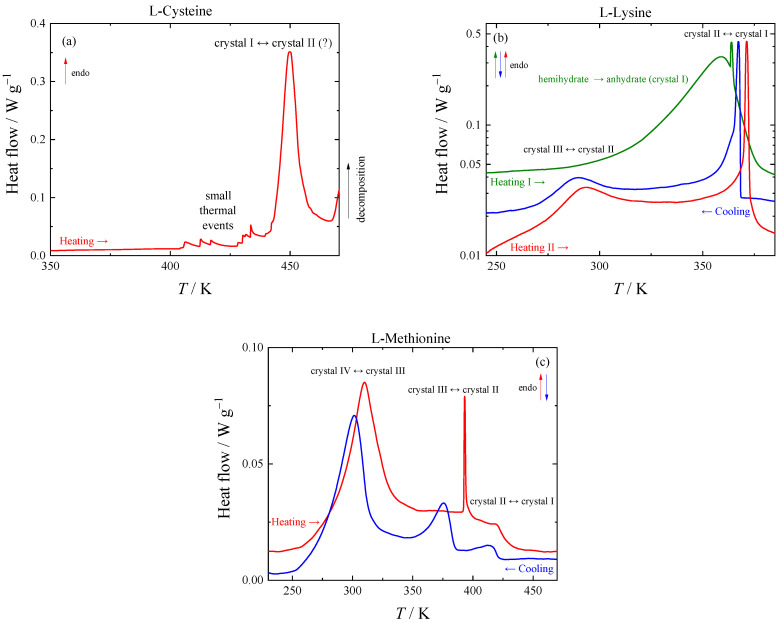
Thermograms obtained using DSC TA Q1000 for (**a**): L-cysteine, (**b**) L-lysine (logarithmic scale), (**c**): L-methionine. Line colors distinguish individual thermal cycles.

**Figure 3 molecules-28-00451-f003:**
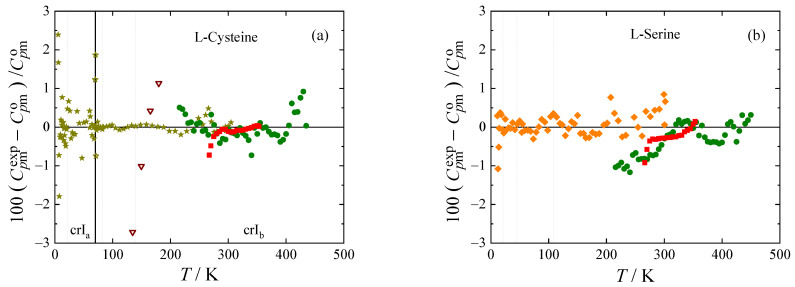
Relative deviations $100(C_{pm}^{\exp}-C_{pm}^{o})/C_{pm}^{o}$ of individual experimental heat capacities $C_{pm}^{\exp}$ from values $C_{pm}^{o}$ calculated by means of Equations (1) and (2) with parameters from Table 5. (**a**) L-cysteine, (**b**) L-serine, (**c**) L-threonine, (**d**) L-lysine, and (**e**,**f**) L-methionine. Blue ▲, this work (relaxation calorimetry); red ∎, this work (Tian-Calvet calorimetry); green ⬤, this work (power compensation DSC); olive ★, Paukov et al. [30]; brown ▽, Huffman and Ellis [46]; orange ◆, Hutchens et al. [34,47]; purple ⬠, Lukyanova et al. [12]. Dotted vertical lines mark knot temperatures *T_i_,* thick vertical lines phase transition temperatures. Data points represented by filled symbols were used to obtain parameters of Equations (1) and (2) in Table 5.

**Figure 4 molecules-28-00451-f004:**
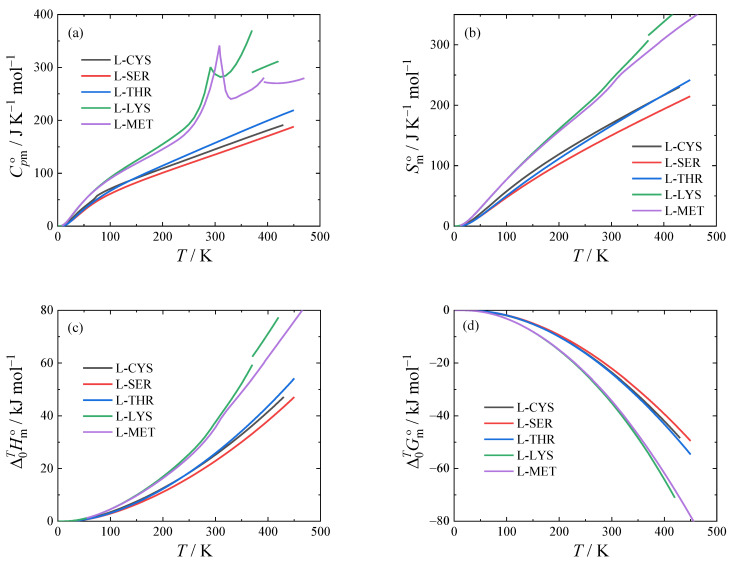
Standard molar thermodynamic functions of stable forms at *p* = 0.1 MPa. (**a**) isobaric heat capacity, (**b**) entropy, (**c**) enthalpy, and (**d**) Gibbs energy.

**Table 2 molecules-28-00451-t002:** Normal-Pressure Crystal Structures of Amino Acids Studied in this work.

Compound	Phase	Refcode ^a^	*Z*	Space Group	Ref. ^b^
L-cysteine	I_a_	LCYSTN22	4	*P*2_1_2_1_2_1_	[24]
	I_b_	LCYSTN12	4	*P*2_1_2_1_2_1_	[25]
L-serine	I	LSERIN01	4	*P*2_1_2_1_2_1_	[26]
L-threonine	I	LTHREO	4	*P*2_1_2_1_2_1_	[27]
L-lysine	HH ^c^	UPUKUN^c^	4	*C*2	[9]
	III/II ^d^	CUFFUG^d^	4	*P*2_1_	[10]
L-methionine	IV	LMETON10	4	*P*2_1_	[28]
	III	LMETON13	4	*P*2_1_	[29]
	II	LMETON14	4	*P*2_1_	[29]

^a^ Identifier in the Cambridge Structural Database (CSD). ^b^ Reference in which crystal structure parameters with a given Refcode were determined. ^c^ Based on the comparison of the obtained diffractogram with that for anhydrous L-lysine and L-lysine hemihydrate (HH), the “as received” sample was identified as L-lysine hemihydrate. ^d^ Reference [10] does not contain information about temperature. Since the phase transition III-II is in the vicinity of the room temperature, refcode CUFFUG may correspond to either of the structures.

**Table 3 molecules-28-00451-t003:** Phase Transitions of L-Cysteine, L-Lysine, and L-Methionine.

Reference	*T*_transition_/K	Δ*H*_transition_/kJ·mol^−1^	Method	Notes
L-cysteine, I_a_ → I_b_				
Paukov et al. [30]	70		adiabatic	
L-cysteine, I_b_ → II				
This work	444 ± 3 ^a^	3.3 ± 0.5	DSC	onset, 1st order
L-lysine, III → II				
This work	(292 ± 1)	(1.2) ^b^	DSC	top, 2nd order
This work	291.1	0	*C_p_* correlation	top, 2nd order
L-lysine, II → I				
This work	370.3 ± 0.5	3.1 ± 0.5	DSC	onset, 1st order
L-methionine (unclear phase transition, see text)				
Lima Jr et al. [31]	220		relaxation	*C_p_* anomaly
L-methionine, IV → III				
Guinet et al. [32]	260–320		Tian-Calvet	broad
Lima Jr et al. [31]	307	7.7–8.0	DSC	peak top
Grunenberg et al. [33]	307	1.98	DSC	
Hutchens et al. [34]	306	0	adiabatic	
Sabbah and Minadakis [35]	258–361	2.7	DTA	
Roux et al. [36]	308	2.3	DSC	
Görbitz et al. [29]	309		DSC	
This work	308.4 ± 0.3	0	*C_p_* correlation	top, lambda
This work	(310 ± 1)	(3.5) ^b^	DSC	top, lambda
L-methionine, III → II				
Guinet et al. [32]	395		Tian-Calvet	sharp
Grunenberg et al. [33]	393	1.15	DSC	
Sabbah and Minadakis [35]	391–403	0.21	DTA	
Roux et al. [36]	395	0.23	DSC	
Görbitz et al. [29]	393		DSC	
This work	393 ± 1	0.13 ± 0.02	DSC	onset, 1st order
L-methionine, II → I				
Sabbah and Minadakis [35]	408–430	0.09	DTA	
Roux et al. [36]	421	0.10	DSC	
Görbitz et al. [29]	424		DSC	
This work	421 ± 1	0.04 ± 0.01	DSC	top, 1st order

^a^ Correspondence of this thermal event to phase transition I→II is suggested but not experimentally confirmed. Probably overheated phase transition, small thermal events were observed starting from 400 K as seen in Figure 2a. ^b^ Phase transition was considered to be of second order and thus associated with no latent heat. The value of apparent phase-transition enthalpy is listed only for comparison with other DSC studies and was not considered when developing the thermodynamic description.

**Table 4 molecules-28-00451-t004:** Overview of the Literature Crystal Heat Capacities of L-Cysteine, L-Serine, L-Threonine, L-Lysine, and L-Methionine.

Reference ^a^	*N* ^b^	(*T*_min_–*T*_max_)/K	$100u_{r}(C_{pm}^{o}){\;}^{c}$	Method
L-cysteine				
Huffman and Ellis [46]	17	85–298	1.0	adiabatic
**Paukov et al.** [30]	**87 *+*** *3* ^d^	**6–304**	**0.1**	**adiabatic**
**This Work**	**19**	**267–353**	**1.0**	**Tian-Calvet**
**This Work**	**44**	**216–430**	**3.0**	**PC DSC**
L-serine				
**Hutchens et al.** [47]	**61**	**11–302**	**~0.4** [48]	**adiabatic**
**This Work**	**19**	**266–353**	**1.0**	**Tian-Calvet**
**This Work**	**48**	**216–450**	**3.0**	**PC DSC**
L-threonine				
Lukyanova et al. [12]	42S	10–370	0.2	adiabatic
**This Work**	**15**	**266–332**	**1.0**	**Tian-Calvet**
**This Work**	**126**	**2–303**	^e^	**Relaxation**
**This Work**	**48**	**216–450**	**3.0**	**PC DSC**
L-lysine				
**This Work**	**26** *+ 1* ^d^	**232–360**	**1.0**	**Tian-Calvet**
**This Work**	**118**	**2–303**	^e^	**Relaxation**
**This Work**	**38** *+ 4* ^d^	**216–420**	**3.0**	**PC DSC**
L-methionine				
**Hutchens et al.** [34]	**70** *+ 1* ^d^	**11–305**	**~0.4** [48]	**adiabatic**
Hutchens et al. [34]	14	307–348	nosp. ^f^	adiabatic
Lima Jr et al. [31]	G	5–390	nosp. ^f^	relaxation
**This Work**	**19**	**266–353**	**1.0**	**Tian-Calvet**
**This Work**	**48** *+ 4* ^d^	**216–470**	**3.0**	**PC DSC**

^a^ The data from references written in bold were fitted to equations 1 and 2. ^b^ *N* = number of data points. ‘S’ stands for smoothed data, and ‘G’ stands for graphical form only. ^c^ *u*_r_($C_{pm}^{o}$) stands for relative uncertainty in heat capacity, as stated by the authors. ^d^ Points not considered during the correlation, typically because of the vicinity of a phase transition. ^e^ For specification of *u*_r_($C_{pm}^{o}$) of PPMS using thermal relaxation measurement technique, see Section 3.4. ^f^ nosp. stands for not specified.

**Table 5 molecules-28-00451-t005:** Parameters of Equations (1) and (2) for Crystal Heat Capacities in J K^−1^ mol^−1^.

	*a* _i_	*b* _i_	*c* _i_	*d* _i_	*T*_i_/K	*T*_i+1_/K	*N* ^a^	*s* _r_ ^b^
	L-cysteine			*b*^c^ = 0.15				
crystal I_a_	−4.97256 × 10^−4^	3.50119 × 10^−2^	−8.41444 × 10^−1^	1.40275 × 10^+1^	0	22	21	0.87
	−2.56251 × 10^−5^	2.19297 × 10^−3^	−2.29383 × 10^−2^	7.16667 × 10^+0^	22	70	19	0.35
crystal I_b_	−6.72981 × 10^−5^	2.37420 × 10^−3^	1.36686 × 10^−2^	8.07220 × 10^+0^	70	82	11	0.06
	−6.08764 × 10^−7^	−4.85326 × 10^−5^	4.15766 × 10^−2^	8.46182 × 10^+0^	82	140	11	0.04
	1.61689 × 10^−7^	−1.54458 × 10^−4^	2.98032 × 10^−2^	1.05912 × 10^+1^	140	435	89	0.26
	L-serine			*b*^c^ = 0.16				
crystal I	−5.06946 × 10^−4^	3.32765 × 10^−2^	−7.50659 × 10^−1^	1.38501 × 10^+1^	0	21	7	0.64
	−1.46033 × 10^−5^	1.33889 × 10^−3^	−2.37366 × 10^−2^	8.06633 × 10^+0^	21	45	7	0.11
	−2.18531 × 10^−6^	2.87454 × 10^−4^	1.52956 × 10^−2^	8.06598 × 10^+0^	45	108	14	0.16
	7.60120 × 10^−8^	−1.25569 × 10^−4^	2.54944 × 10^−2^	9.62408 × 10^+0^	108	450	100	0.45
	L-threonine			*b*^c^ = 0.15				
crystal I	3.20725 × 10^−3^	−8.38463 × 10^−2^	2.72915 × 10^−1^	1.37048 × 10^+1^	0	11	20	1.88
	−4.44502 × 10^−4^	2.19929 × 10^−2^	−4.07471 × 10^−1^	1.08303 × 10^+1^	11	25	14	0.30
	−4.45102 × 10^−5^	3.32385 × 10^−3^	−5.30363 × 10^−2^	8.21661 × 10^+0^	25	50	12	0.20
	−3.66263 × 10^−7^	−1.44160 × 10^−5^	2.96995 × 10^−2^	8.27263 × 10^+0^	50	156	37	0.29
	1.34723 × 10^−7^	−1.30888 × 10^−4^	1.42973 × 10^−2^	1.08226 × 10^+1^	156	450	106	0.37
	L-lysine			*b*^c^ = 0.12				
crystal III	−1.05842 × 10^−2^	2.00379 × 10^−1^	−1.38437 × 10^+0^	1.02633 × 10^+1^	0	6	14	0.62
	−2.15249 × 10^−4^	9.86222 × 10^−3^	−1.22924 × 10^−1^	6.88452 × 10^+0^	6	20	16	0.29
	−9.32527 × 10^−6^	8.21784 × 10^−4^	2.66518 × 10^−2^	6.50594 × 10^+0^	20	51	18	0.17
	−3.48146 × 10^−7^	−4.54662 × 10^−5^	5.07176 × 10^−2^	7.84407 × 10^+0^	51	234	61	0.62
	−2.21751 × 10^−5^	−2.36598 × 10^−4^	−9.00177 × 10^−4^	1.34692 × 10^+1^	234	291.1	37	0.82
crystal II	1.03861 × 10^−5^	−2.66677 × 10^−3^	1.28741 × 10^−1^	8.52650 × 10^+0^	291.1	370.3	26	0.58
crystal I	0	0	1.91791 × 10^−3^	1.14804 × 10^+1^	370.3	420	10	0.29
	L-methionine			*b*^c^ = 0.10				
crystal IV	−3.82533 × 10^−5^	4.58795 × 10^−3^	−1.10007 × 10^−1^	7.55129 × 10^+0^	0	40	13	0.55
	−4.13881 × 10^−7^	−2.44633 × 10^−6^	7.34130 × 10^−2^	8.04351 × 10^+0^	40	216	35	0.13
	−8.39290 × 10^−6^	−2.20975 × 10^−4^	3.40908 × 10^−2^	1.86320 × 10^+1^	216	308.4	50	0.80
crystal III	1.94174 × 10^−4^	−1.89543 × 10^−2^	6.32364 × 10^−1^	1.33080 × 10^+1^	308.4	342	14	0.94
	−1.68487 × 10^−5^	6.18467 × 10^−4^	1.62806 × 10^−2^	2.05224 × 10^+1^	342	393	13	0.40
crystal II + I	0	−3.01298 × 10^−4^	6.92732 × 10^−2^	2.13260 × 10^+1^	393	470	12	0.20

^a^ *N* stands for the number of experimental data points in a given temperature interval used for correlation. ^b^ $s_{r}=100{\left\{\sum_{i=1}^{n}{\left[\left(C_{pm}^{\exp}-C_{pm}^{o}\right)/C_{pm}^{o}\right]}_{i}^{2}/\left(N-m\right)\right\}}^{1/2}$, where $C_{pm}^{\exp}$ and $C_{pm}^{o}$ is experimental and calculated (Equations (1) and (2)) heat capacity, *N* is the number of fitted data points, and *m* is the number of independent adjustable parameters. ^c^ parameter *b* from Equation (1).

**Table 6 molecules-28-00451-t006:** Standard Thermodynamic Functions of Stable Forms of L-Cysteine, L-Serine, L-Threonine, L-Lysine, and L-Methionine at *p* = 0.1 MPa and *T* = 298.15 K ^a^.

Compound	$C_{pm}^{o}/J\cdot K^{-1}\cdot {\mathrm{mol}}^{-1}$	$S_{m}^{o}/J\cdot K^{-1}\cdot {\mathrm{mol}}^{-1}$	$\Delta_{0}^{T}H_{m}^{o}/\mathrm{kJ}\cdot {\mathrm{mol}}^{-1}$	$\Delta_{0}^{T}G_{m}^{o}/\mathrm{kJ}\cdot {\mathrm{mol}}^{-1}$
L-cysteine	144.6	169.0	24.96	−23.28
L-serine	134.7	149.1	22.64	−21.82
L-threonine	156.5	165.1	25.54	−23.69
L-lysine ^b^	289.2	240.7	37.09	−34.69
L-methionine ^b^	288.9	231.5	35.09	−33.91

^a^ The combined expanded uncertainty of heat capacity $U_{c}(C_{pm}^{o})$ as well as of all calculated thermodynamic values (with 0.95 level of confidence, *k* = 2) is *U*_c_(*X*) = 0.01 *X* at 298.15 K, where *X* represents the heat capacity or the thermodynamic property. Values are reported with one digit more than is justified by the experimental uncertainty to avoid round-off errors in calculations based on these results. ^b^ Second-order phase transitions lie close to 298.15 K. These are not associated with a step change of enthalpy and entropy but manifest by an increased heat capacity.

**Table 7 molecules-28-00451-t007:** Sample Description.

Compound	CAS RN	Supplier	Mole Fraction Purity ^a^	Purity Method ^a^
L-cysteine	52-90-4	Sigma-Aldrich	0.993	titration redox
L-serine	56-45-1	Fisher Scientific	0.998	nosp ^b^
L-threonine	72-19-5	Sigma-Aldrich	1.000	TLC ^c^
L-lysine	56-87-1	Sigma-Aldrich	1.000	TLC ^c^
L-methionine	63-68-3	Sigma-Aldrich	1.000	titration HClO_4_

^a^ From certificate of analysis supplied by the manufacturer. ^b^ nosp stands for not specified. ^c^ TLC stands for thin-layer chromatography.

## Data Availability

The data presented in this study are available in the Appendix A.

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
