# Peer review of "Heat Capacities of L-Cysteine, L-Serine, L-Threonine, L-Lysine, and L-Methionine"

_molecules, 2023, doi:10.3390/molecules28010451_

Round 1

Reviewer 1 Report

Excellent work in the field of thermodynamic data and correlation creation for amino acids.  There are just a few comments mentioned in the file. 

From the observations, it seems that data significantly varies depending on the measurement techniques/instruments concerned. How to overcome this discrepancy?

Author Response

Reviewer 1:

Comments and Suggestions for Authors

Excellent work in the field of thermodynamic data and correlation creation for amino acids.  There are just a few comments mentioned in the file. 

Answer: We thank Reviewer 1 for highly positive evaluation of our work.

From the observations, it seems that data significantly varies depending on the measurement techniques/instruments concerned. How to overcome this discrepancy?

Answer: This discrepancy can be overcome only for new measurements (which can be repeated if necessary). It is however difficult to comment discrepancies between data from two different literature sources, sometimes it is impossible to find what is the origin of discrepancies (e.g. when data from given laboratory are in good agreement with data from other laboratories for given group of compounds but differ for other compounds in consequent paper).
Typically, discrepancies can be resolved by new measurements, preferably in separate laboratories or by several techniques.

Comments mentioned in the file:

  • Page 1 line 22 addition of “compared“ suggested

Action: Accepted, the sentence was extended and its final version is

Crystal heat capacities of all five amino acids were measured by Tian–Calvet calorimetry in the temperature interval (262 – 358) K and by power compensation DSC in the temperature interval from 215 K to over 420 K. Experimental values of this work were compared and combined with the literature data obtained with adiabatic calorimetry.

  • Page 1 line 45, deletion of “e.g.“ suggested
    Action: Accepted, “e.g.“ was deleted and the final version of this sentence is

 Significance and many other aspects of title as well as other amino acids are summarized in reviews [6,8].

  • Page 2 line 63 “s“ was missing in the word suggests
    Action: Corrected, suggest was changed to suggests.

  • Page 12., Fig 3 a Reviewer 1 added a comment “ This shows that measurement techniques are also responsible for significant variations.”
    Answer: We agree with the reviewer, different techniques are associated with different uncertainties. However, the differences of values obtained by different techniques fall within uncertainty assigned to respective techniques (those uncertainties are described in Section 3.4. Heat capacity measurements on pages 14 and 15).  Exceptions from the above statements were observed for some literature datasets as discussed on lines 274 to 313 (pages 10 and 11).
    Action: no changes were made in the manuscript.

  • Page 13, Figure 4. Reviewer 1 asks “Why break“ (on the temperature dependence of heat capacities).
    Answer: Such breaks are natural for all first order phase transition. Since thermodynamic potentials (H, S, U) exhibit step change during a first order phase transition, there is not any requirement toward their derivatives (heat capacity) at the temperature of the phase transition.
    Action: no changes were made in the manuscript.

Reviewer 2 Report

The manuscript entitled "Heat capacities of L-cysteine, L-serine, L-threonine, L-lysine, 2 and L-methionine" refers to determination, measurement or confirmation the reliable thermodynamic data of 5  aminoacids L-cysteine, L-serine, L-threonine, L-lysine, and L-methionine. First, authors confirmed the crystal structures of the tested aminoacids  by XRD method and next they investigated the polymorphic forms of each tested aminoacids by DSC method.

The decomposition temperature and the way of decomposition of aminoacids were also tested by TGA.

 Low temperature heat capacities of aminoacids were measured by  the relaxation calorimetry and the reference heat capacities and thermodynamic functions of the tested aminoacids in the range of temperature near 0 K to 420 K were established.

Authors compared and discussed  their results with the literature available data.

The work is very well-done, valuable and can be published as it is.

I have only one suggestion. Maybe TMDSC would be useful for polymorphic forms analysis of the tested aminoacids.

Author Response

Reviewer 2:

Comments and Suggestions for Authors

The manuscript entitled "Heat capacities of L-cysteine, L-serine, L-threonine, L-lysine, 2 and L-methionine" refers to determination, measurement or confirmation the reliable thermodynamic data of 5  aminoacids L-cysteine, L-serine, L-threonine, L-lysine, and L-methionine. First, authors confirmed the crystal structures of the tested aminoacids  by XRD method and next they investigated the polymorphic forms of each tested aminoacids by DSC method.

The decomposition temperature and the way of decomposition of aminoacids were also tested by TGA.

 Low temperature heat capacities of aminoacids were measured by  the relaxation calorimetry and the reference heat capacities and thermodynamic functions of the tested aminoacids in the range of temperature near 0 K to 420 K were established.

Authors compared and discussed  their results with the literature available data.

The work is very well-done, valuable and can be published as it is.

Answer: We thank Reviewer 2 for highly positive evaluation of our work.

I have only one suggestion. Maybe TMDSC would be useful for polymorphic forms analysis of the tested aminoacids.

Answer: Thank you for this suggestion. We have been using used MDSC for study of glass transitions so far. In our future work, we will certainly test this technique for phase transitions, too. Following the suggestion of Reviewer 2, we have found recent interesting review by Leyva-Porras et al., Application of Differential Scanning Calorimetry (DSC) and Modulated Differential Scanning Calorimetry (MDSC) in Food and Drug Industries, Polymers 2019, 12, 5; doi:10.3390/polym12010005 .